# Vitamin D deficiency and SARS-CoV-2 infection: A retrospective case–Control study with big-data analysis covering March 2020 to March 2021

**Marta Neira Álvarez**[1], **Gema Navarro Jiménez**[2]*, **Noemi Anguita Sánchez**[3]*, **Mª del Mar Bermejo Olano**[2], **Rocio Queipo**[4], **María Benavent Núñez**[5], **Alejandro Parralejo Jimenez**[5], **Guillermo López Yepes**[5], **Carmen Sáez Nieto**[6]

1 Department of Geriatrics, Infanta Sofía and Henares Hospitals Foundation for Biomedical Research and Innovation (FIIB HUIS HHEN), Universidad Europea Madrid, Madrid, Spain, 2 Department of Internal Medicine, Infanta Sofía University Hospital, Infanta Sofía and Henares Hospitals Foundation for Biomedical Research and Innovation (FIIB HUIS HHEN), Madrid, Spain, 3 Department of Geriatrics, Zendal Hospital, Madrid, Spain, 4 Dept. of Medicine Universidad Europea, Madrid, Spain, 5 Medsavana, Madrid, Spain, 6 Department of Hospital Medicine, Clínico University Hospital, Madrid, Spain

* noemi.anguita@salud.madrid.org (NAS); gnavarroj@salud.madrid.org (GNJ)

## Abstract

### Background

Vitamin D may have immunomodulatory functions, and might therefore play a role in the pathogenesis of acute respiratory syndrome coronavirus 2 (SARS-CoV-2) infection. However, no conclusive evidence exists regarding its impact on the prevalence of this infection, the associated course of disease, or prognosis.

### Objective

To study the association between SARS-CoV-2 infection and vitamin D deficiency in patients attending a tertiary university hospital, and to examine the clinical course of infection and prognosis for these patients.

### Methods

This non-interventional, retrospective study, which involved big-data analysis and employed artificial intelligence to capture data from free text in the electronic health records of patients diagnosed with SARS-CoV-2, was undertaken at a tertiary university hospital in Madrid, Spain, between March 2020 and March 2021. The variables recorded were vitamin D deficiency, sociodemographic and clinical characteristics, course of disease, and prognosis.

### Results

Of the 143,157 patients analysed, 36,261 had SARS-CoV-2 infection (25.33%) during the study period, among whom 2,588 (7.14%) had a vitamin D deficiency. Among these latter patients, women (OR 1.45 [95%CI 1.33–1.57]), adults over 80 years of age (OR 2.63 [95%

**Data Availability Statement:** dataset is published in open access in zenodo: https://doi.org/10.5281/zenodo.7053208.

**Funding:** The authors received no specific funding for this work.

**Competing interests:** The authors have declared that no competing interests exist.

CI 2.38–2.91]), people living in nursing homes (OR 2.88 [95%CI 2.95–3.45]), and patients with walking dependence (OR 3.45 [95%CI 2.85–4.26]) appeared in higher proportion. After adjusting for confounding factors, a higher proportion of subjects with SARS-CoV-2 plus vitamin D deficiency required hospitalisation (OR 1.38 [95%CI 1.26–1.51]), and had a longer mean hospital stay (3.94 compared to 2.19 days in those with normal levels; P = 0.02).

## Conclusion

A low serum 25(OH) vitamin D concentration in patients with SARS-CoV-2 infection is significantly associated with a greater risk of hospitalisation and a longer hospital stay. Among such patients, higher proportions of institutionalised and dependent people over 80 years of age were detected.

## Introduction

Vitamin D is a steroid, classically associated with bone metabolism. However, in recent years, important extra-osseous regulatory functions have been described for this vitamin, including an immunomodulatory function [1]. Vitamin D seems to activate the innate immune system by stimulating antigen presentation to macrophages, by activating neutrophils and T cells, and by reducing the cytokine storm at the site of infection [2]. Several hypotheses have been advanced regarding the role of vitamin D in acute respiratory syndrome coronavirus 2 (SARS-CoV-2) infection. Some studies show a higher risk of respiratory infection in patients with vitamin D deficiency, while others show a higher incidence of SARS-CoV-2 infection and greater mortality in countries where vitamin D deficiency is prevalent [3–6]. However, other authors report inconsistent results regarding vitamin D status and diagnosis with SARS-CoV-2 infection, the lack of any association with hospitalization or mortality, and no difference in the length of hospital stay when patients with moderate to severe SARS-CoV-2 infection are provided with oral supplements of vitamin D [7].

The evidence supporting a role for vitamin D in SARS-CoV-2 infection therefore remains inconclusive.

A high prevalence of vitamin D deficiency has been noted for the Spanish population, particularly among the elderly and institutionalised [8–10].

Members of the latter groups are commonly little exposed to the sun, and often have nutritional disorders or chronic kidney disease—all factors associated with vitamin D deficiency. In addition, the confinement imposed for two years in an attempt to control the spread of SARS-CoV-2 may have increased vitamin D deficiency among such persons, with further reductions in their exposure to the sun and the worsening of underlying nutritional disorders. If vitamin D deficiency is associated with a greater severity of symptoms and a more serious course of disease, such elderly populations might be the most vulnerable.

The main aim of the present work was to study the prevalence of vitamin D deficiency among patients with SARS-CoV-2 infection admitted to a tertiary hospital during the period March 2020 to March 2021. In addition, the study aimed to: 1) determine the characteristics of patients with SARS-CoV-2 plus vitamin D deficiency (sociodemographic characteristics, comorbidities, vitamin D treatments, level of walking autonomy, place of residence); 2) compare the clinical course of disease (need for hospital admission, duration of hospital stay, need for intensive care, and mortality) in patients with SARS-CoV-2 infection plus a deficient or

normal vitamin D status; and 3) to examine these differences during the three sub-periods of maximum incidence of SARS-CoV-2 infection, i.e., March 2020 to June 2020, August 2020 to October 2020, and January 2021 to March 2021.

## Materials and methods

This non-interventional, retrospective study, which involved big-data analysis and employed artificial intelligence to capture data from free text in the electronic health records (EHRs) of patients diagnosed with SARS-CoV-2, was undertaken at a tertiary university hospital in Madrid, Spain. The study was approved by the Ethics and Research Committee of the University Hospital of Getafe (Ref Number: CEIm21/40) on the 31st of September 2021.

### Sources of information and search strategies

Clinical data were collected for 143,157 patients diagnosed with SARS-CoV-2 between March 2020 and March 2021 (detected from among 203,4921 EHRs held at the Infanta Sofía University Hospital (Madrid, Spain). Data were collected from all available departments, including the hospital in-patient and out-patient departments, emergency room, and laboratory. Information was extracted from the EHRs using natural language processing (NLP) and artificial intelligence (AI) techniques. Savana Manager software, used to analyse the free text clinical information in the patients' EHRs. This software can interpret all EHR content, regardless of the electronic system in which it is required to operate. Importantly, it can capture numerical values and clinical notes and transform them into accessible variables, allowing for big-data analyses. The data extraction process has four distinct phases for transferring and aggregating data into a study database:

1. *Acquisition*. This is the responsibility of the participating centre, and is performed in close collaboration with the Medsavana company's technical staff. The data are 'pseudonymized' using the software's "anonymizer", and then transferred to the Medsavana server using a secure file transfer protocol.

2. *Integration*. In this phase, data are entered into a study database.

3. *NLP*. Savana EHRead technology then uses NLP techniques to analyse and extract unstructured, free text information found in the patient EHRs. This creates structured, clinical data from this unstructured information. No information can be later traced to individual patients. The terminology used by the software is based on multiple sources such as SNOMED CT [11], which includes medical codes, concepts, synonyms, and definitions regarding symptoms, diagnoses, body structures and substances commonly used in clinical documentation.

4. *Validation*. This entails virifiying de system's accuracy in identifiying records that contain mentions of vitamin D deficiency, SARS-CoV2-2 and related variables. For this human-annotated corpus of EHRs known as an "annotation gold standard" is created for comparison. In the present work the SampLe Calculator for Evaluation (SLiCE) tool was used to determine size of the corpus required (with a confidence level of 95% and interval widths of 5% based on the prevalence of mentions of on the prevalenceof mentions of the primary research variables (vitamin D deficiency, SARS-CoV-2). Although the number of EHRs required to adequately capture linguistic occurrences in order to generate reliable performance measurements is still a matter of debate, this methodology provides a robust estimate of precision (P) and recall (R) values. The annotation gold standard was constructed using 98 records annotated by two designated researchers ("the annotators") using the

Savana Evaluation Tool to manually correct the EHRead-selected variables. The Inter-Annotator Agreement (IAA) was determined using the F1-Score. A third researcher resolved the differences when the annotators disagreed.

The generated gold standard corpus was then used to assess performance of the EHRead technology, calculating:

- Precision = tp/(tp+fp). This shows how well the system recalls important therapeutic concepts.

- Recall = tp/(tp+fn). This represents the volume of data that the system retrieves.

- F1-Score = (2 x precision x recall/(precision + recall). This provides an indication of retrieval performance.

In each case, tp is the total number of true positives (i.e., correctly retrieved records), fn is the total number of false negatives (i.e., wrongly omitted records), and fp is the total number of false positives (i.e., records incorrectly retrieved) [12].

The linguistic evaluation of the variable 'SARS-CoV-2' yielded accuracy, recall and F-scores of 0.61, 0.87, and 0.72, respectively. The Interannotator Agreement was 0.70. Other primary variables such as hypovitaminosis D and malnutrition yielded F-scores of >0.70.

## Study population

All the EHRs used in the present work belonged to patients attending the University Infanta Sofía Hospital (Madrid, Spain) between March 2020 and March 2021. All had a diagnosis of SARS-CoV-2 infection confirmed by either PCR or an antigen test at the above hospital.

## Study variables

The following study variables were categorized as follows:

- *SARS-CoV-2 infection*: positive or negative for SARS-CoV-2 infection.

- *Vitamin D deficiency*: positive or negative for vitamin D deficiency defined as vitamin D deficiency, hypovitaminosis D or vitamin D blood levels <30 mg/dl in the EHR (detected before or during the SARS-CoV-2 infection).

- *Comorbidities*: presence of hypertension, diabetes, obesity, malnutrition, dementia, chronic kidney disease, liver disease, malabsorption disorders.

- *Clinical course of SARS-CoV-2 infection*: hospital admission, admission to the intensive care unit (ICU), hospital stay (in days), and mortality.

- *Walking independence*: autonomous, requiring technical assistance, total dependence.

- *Vitamin D treatments*: whether patients received vitamin D supplementation before or during the SARS-CoV-2 infection

- *Time of episode (study sub-period)*: March 2020 to June 2020, August 2020 to October 2020, and January 2021 to March 2021.

## Data reporting

The data that support the present findings are available online on the ZENODO open access site: https://doi.org/10.5281/zenodo.7053208.

## Statistical analysis

Unless otherwise indicated, qualitative variables were expressed as absolute frequencies and percentages, and quantitative variables as mean±SD. Differences between numerical variables were examined using the t-test for independent samples. The Chi-squared test was used to examine the relative distribution of patients assigned to different categories of qualitative variables. Significance was set at P<0.05.

Three multiple logistic regressions were performed to assess differences in hospitalisation rates, ICU admission and mortality between groups, adjusting for covariates (age, obesity, malnutrition, living in a nursing home, and SARS-CoV-2 infection). The results were expressed as odds ratios (OR) with 95%CI. All analyses were conducted using SPSS software v.25.0 (IBM, Armonk, NY, USA).

## Ethics

This work adheres to the 'Strengthening the Reporting of Observational Studies in Epidemiology (STROBE) [13] guidelines for reporting observational studies. The study was conducted following all legal and regulatory requirements, and met the research practice requirements described in the 'International Conference on Harmonisation' guidelines. It also adheres to the Declaration of Helsinki (latest edition), the guidelines for good pharmacoepidemiological practice, and all local regulations. Given the retrospective and observational nature of the study, prescribing habits and patient assignment to specific therapeutic strategies were determined solely by the physicians concerned. Since the study involved data in EHRs, it entailed no physical risk for any participant; informed consent was therefore not required. Indeed, given the use of aggregated data it would have been impossible to identify patients in order to seek any consent. Data protection was ensured in line with the European Data Protection Authority's code of good practice regarding big-data projects, and the European General Data Protection Regulation.

# Results

Of the 143,157 patient records analysed, 6,666 were excluded owing to a missing data. 36,261 subjects had SARS-CoV-2 infection (25.33%); 2588 of them also had a vitamin D deficiency (7.14%).

## Characteristics of the study population

Table 1 shows the general characteristics of the study subjects with SARS-CoV-2 infection, with and without vitamin D deficiency. Among the patients with SARS-CoV-2 plus vitamin D deficiency, a higher proportion of women (OR 1.45 [95%CI 1.33–1.57]), of adults between 65 and 80 years of age (OR 1.86 [95%CI 1.70–2.0]), and of people over 80 years of age (OR 2.63 [95%CI 2.38–2.91]) was noted, along with a higher proportion of people living in nursing homes (OR 2.88 [95%CI 2.95–3.45]), and with walking dependence (OR 3.45 [95%CI 2.85–4.26]). In addition, higher proportions of patients with malnutrition, hypertension and obesity were noted; these patients more frequently received vitamin D supplements.

## Severity of SARS-CoV-2 infection

Patients with SARS-CoV-2 plus vitamin D deficiency more often required hospitalisation (OR 2.41 [95%CI 2.22-2-61]), treatment in the ICU (OR 2.22 [95% CI 1.64–3.02]), and had a higher mortality rate (OR 1.82 [95%CI 1.66–2.01]). However, after risk-adjustment, only the hospitalisation rate remained significant (OR 1.38 [95%CI 1.26–1.51]). Patients with low vitamin D levels also had a longer mean hospital stay (3.94 compared to 2.29 days in those with normal levels; P = 0.02) (**Tables 2 and 3**).

**Table 1. General characteristics of subjects with SARS-CoV-2 infection with and without vitamin D deficiency.**

| | Patients with SARS-CoV-2 infection | SARS-CoV-2 infection plus vitamin D deficiency | | SARS-CoV-2 infection without vitamin D deficiency | | OR (95%CI) | P value |
|---|---|---|---|---|---|---|---|
| **Patients n (%)p** | 36,261 | 2,588 | (7.1) | 33,673 | (92.8) | | |
| **Sex n, (%)** | | | | | | | |
| Women | 20,170 | 1,662 | (64.2) | 18,508 | (54.9) | 1.45 (1.33–1.58) | <0.001 |
| Men | 15,525 | 905 | (34.9) | 14,620 | (43.4) | 0.69 (0.63 0.75) | <0.001 |
| **Mean age mean (SD)** | 55.19 (23.5) | 65.48 (20.5) | | 55.96 (23.1) | | 0.04 | |
| <18 years (n, %) | 4,677 | 106 | (4.1) | 4,571 | (13.5) | 1 (0.22–0.33) | <0.001 |
| 18–64 years (n, %) | 20,752 | 1,185 | (45.7) | 19,567 | (58.1) | 1 (0.56–0.66) | <0.001 |
| 65–80 years (n,%) | 6,591 | 730 | (28.2) | 5,861 | (17.4) | 1 (1.70–2.040) | <0.001 |
| > 80 years (n,%) | 3,867 | 575 | (22.2) | 3,292 | (9.7) | 1 (2.39–2.91) | <0.001 |
| **Place of living n (%)** | | | | | | | |
| Home | 35,081 | 2,379 | (91.9) | 32,702 | (97.1) | 0.37 (0.32–0.44) | <0.001 |
| Nursing home | 1,180 | 209 | (8.1) | 971 | (2.8) | 2.96 (2.53–3.46) | <0.001 |
| **Comorbidities n (%)** | | | | | | | |
| Hypertension | 20,587 | 1,994 | (77.1) | 18,593 | (55.2) | 2.72 (2.48–2.99) | <0.001 |
| Diabetes | 8,379 | 435 | (16.8) | 7,944 | (23.6) | 0.65 (0.59–0.73) | <0.001 |
| Chronic kidney disease | 55 | 5 | (0.2) | 50 | (0.1) | 1.30 (0.52–3.27) | 0.57 |
| Obesity | 213 | 23 | (0.9) | 190 | (0.5) | 1.58 (1.02–2.44) | 0.04 |
| Malnutrition | 14 | 4 | (0.1) | 10 | (0.0) | 5.21 (1.63–16.63) | <0.005 |
| Dementia | 3,560 | 59 | (2.3) | 3,501 | (10.4) | 0.20 (1.15–0.26) | <0.001 |
| Liver disease | 1 | 0 | (0.0) | 1 | (0.0) | 0 | 0.78 |
| Malabsorption | 121 | 0 | (0.0) | 121 | (0.3) | 0 | <0.005 |
| Vitamin D treatment D (n,%) | 10,715 | 2,287 | (88.37) | 8,428 | (25.0) | 22,76 (20,14–25,73) | <0,001 |
| **Walking independence (n, %)** | | | | | | | |
| Total independence | 34,451 | 2,228 | (86.1) | 32,223 | (95.6) | 0.28 (0.25–0.31) | <0.001 |
| Needs assistance | 1,197 | 234 | (9.0) | 963 | (2.8) | 3.37 (2.91–3.92) | <0.001 |
| Total dependence | 613 | 126 | (4.8) | 487 | (1.4) | 3.48 (2.85–4.26) | <0.001 |

## Differences between the three periods of maximum incidence

The highest number of SARS-CoV-2 infections was detected in the March to June 2020 sub-period, and the lowest in the January to March 2021 sub-period. SARS-CoV-2 infection plus vitamin D deficiency was most common in this last period (winter). During all three sub-periods, patients with SARS-CoV-2 plus vitamin D deficiency had higher rates of hospitalisation (as they did for the study period as a whole), but only in the March to June 2020 sub-period were longer hospital stays noted (**Table 4**).

**Table 2. Clinical course of SARS-CoV-2 infection.**

| | TOTAL PATIENTS | SARS-CoV-2 PLUS VITAMIN D DEFICIENCY | | SARS-CoV-2 WITHOUT VITAMIN D DEFICIENCY | | OR (95%CI) | P value |
|---|---|---|---|---|---|---|---|
| **TOTAL** | 36,261 | 2,588 | (100.0) | 33,673 | (100.0) | | |
| Hospitalisation | 10,107 | 1,202 | (46.4) | 8,905 | (26.4) | 2.41 (2.22–2.62) | <0.001 |
| Intensive Care | 338 | 49 | (1.8) | 289 | (0.8) | 2.23 (1.64–3.03) | <0.001 |
| Mortality n(%) | 5,482 | 611 | (23.6) | 4,871 | (14.4) | 1.83 (1.66–2.01) | <0.001 |
| Mean hospital stay (SD) | 3.64 (2.3) | 3.94 | (2.3) | 2.19 | (1.9) | | 0.02 |

**Table 3. Clinical course of SARS-CoV-2 infection: Multivariate analysis.**

| Dependent variable: MORTALITY | | | |
|---|---|---|---|
| | P | OR | 95%CI |
| **Age** | 0.000 | 7.87 | 6.93–8.94 |
| **Obesity** | 0.000 | 2.01 | 1.71–2.36 |
| **Mean hospital stay** | 0.000 | 1.09 | 1.05–1.14 |
| **Vitamin D deficency** | 0.332 | 0.89 | 0.70–1.12 |
| **Nursing home** | 0.000 | 2.83 | 2.37–3.39 |
| **SARS-Cov-2 infect** | 0.000 | 6.39 | 5.62–7.26 |
| **Malnutrition** | 0.000 | 2.71 | 2.21–3.30 |
| Dependent variable: INTENSIVE CARE ADMISSION | | | |
| | P | OR | 95%CI |
| **Age** | 0.000 | 0.35 | 0.23–0.54 |
| **Obesity** | 0.000 | 5.02 | 4.07–6.19 |
| **Vitamin D deficency** | 0.982 | 1.00 | 0.69–1.45 |
| **SARS-Cov-2 infect** | 0.000 | 11.76 | 9.29–14.90 |
| **Malnutrition** | 0.000 | 8.64 | 6.35–11.74 |
| Dependent variable: HOSPITALISATION | | | |
| | P | OR | 95%CI |
| **Age** | 0.000 | 1.75 | 1.66–1.85 |
| **Obesity** | 0.000 | 1.69 | 1.59–1.79 |
| **Nursing home** | 0.000 | 2.36 | 2.12–2.62 |
| **Vitamin D deficency** | 0.000 | 1.38 | 1.26–1.51 |
| **SARS-Cov-2 infect** | 0.000 | 3.15 | 3.03–3.27 |
| **Malnutrition** | 0.000 | 2.78 | 2.50–3.08 |

**Table 4. Clinical course of SARS-CoV-2 infection during the three sub-periods of maximum incidence.**

| | TOTAL | TOTAL SARS-CoV-2 INFECTIONS | SARS-CoV-2 PLUS VITAMIN D DEFICIENCY | SARS-CoV-2 WITHOUT VITAMIN D DEFICIENCY | OR (95% CI) | P value |
|---|---|---|---|---|---|---|
| *TOTAL* | 70,147 | 18,555 | 1,563 | 16,992 | | |
| *MARCH TO JUNE 2020* | | | | | | |
| Hospitalisation n (%) | 11,894 | 4,905 | 612(39.1) | 4,293 (25.2) | 1.90 (1.71–2.12) | <0.001 |
| Mean hospital stay (SD) | 2.82 (2.1) | 2.52 (2.1) | 2.8 (2.3) | 1.83 (1.6) | | 0.05 |
| *TOTAL* | 56,818 | 6,950 | 474 | 6,476 | | |
| *AUGUST TO OCTOBER 2020* | | | | | | |
| Hospitalisation n(%) | 10,189 | 2,431 | 283 (59.7) | 2,148 (33.1) | 2.99 (2.47–3.61) | *<0.001* |
| Mean hospital stay (SD) | 2.8 (2.1) | 2.9 (2.3) | 3.2 (2.3) | 2.3 (2.2) | | *0.26* |
| *TOTAL* | 73,321 | 7,892 | 674 | 7,218 | | |
| *JANUARY TO MARCH 2021* | | | | | | |
| Hospitalisation n (%) | 11,587 | 2,972 | 405 (60.1) | 2,567 (35.5) | 2.73 (2.32–3.206) | <0.001 |
| Mean hospital stay (SD) | 2.3 (2.2) | 3.5 (2.3) | 3.3 (2.3) | 3.3 (2.4) | | 0.5 |

## Discussion

This large, retrospective, case control study examines the prevalence of SARS-CoV-2 infection plus vitamin D deficiency during the height of the pandemic, the characteristics of the patients so-affected, and their course of disease.

A significant association was detected between low vitamin D levels and the risk of more serious SARS-CoV-2 disease, a greater risk of the need for hospitalisation, and a longer hospital stay. Significant associations were also found between patients with SARS-CoV-2 plus vitamin D deficiency and sex, age, place of residence, walking dependence and comorbidity factors. Differences were seen between the three sub-periods of maximum incidence in terms of the hospitalisation rate, but not in terms of mean hospital stay. These results are similar to those of other authors who recorded associations between low vitamin D levels and SARS-CoV-2 variables, particularly with respect to the development of severe disease [14–16].

A recent meta-analysis [17] examined the relationship between SARS-CoV-2 infection and vitamin D status. Eight retrospective studies were found that investigated the association between vitamin D levels and SARS-CoV-2 infection, all of which reported that subjects with low vitamin D levels had double the risk. However, most of these studies included unadjusted estimates, rendering it impossible to assess these characteristics in terms of other variables. In addition, 16 studies investigated the association between vitamin D levels and the severity of SARS-CoV-2 infection, reporting double the risk for subjects with low vitamin D levels, even when adjusting for confounding factors (comorbidity, age, BMI and sex). Moreover, supplementation was associated with a significantly lower risk of severe disease in six studies, and with a lower risk of mortality in eight.

Other authors have reported similar results for a Spanish population, noting an increased risk of hospital admission and need for critical care among those with low vitamin D levels, but no increase in mortality [18]. Another retrospective study [19] reported an inverse relationship between vitamin D levels and length of hospital stay for SARS-CoV-2 infection.

The relationship between vitamin D and SARS-CoV-2 infection was initially raised because, prior to the pandemic, a significant association was reported between low serum 25(OH) vitamin D levels and the severity of acute respiratory tract infections in both adults and children [2, 20, 21]. Certainly, vitamin D modulates angiotensin receptor expression in the lungs, stimulates pulmonary surfactant production, reduces hyperinflammatory cytokine storms, and increases levels of regulatory T lymphocytes, which together might explain why vitamin D reduces the incidence of acute respiratory infection and the severity of respiratory tract disease [20]. Vitamin D also has important protective effects on the cardiovascular system, including the improvement of myocardial contractility and anti-thrombotic effects. Moreover, vitamin D deficiency has been associated with diabetes, hypertension and dyslipidemia. It may be, therefore, that this vitamin protects against cardiovascular complications in patients with SARS-CoV-2 infection [22]. Nevertheless, two studies failed to find any protective effects of vitamin D supplements when treating patients hospitalised with SARS-CoV-2, although in both studies the sample was small and heterogeneous, and only patients with severe disease were included [23, 24].

Hypovitaminosis D is commonly observed in the elderly, the malnourished, and those living in nursing homes. The present results show that patients with SARS-CoV-2 plus vitamin D deficiency were usually older, and/or had walking difficulties, and/or suffered chronic diseases associated with vitamin D deficiency such as malnutrition, obesity or chronic kidney disease. This might explain the higher risk of complications such as ICU admission and exitus.

Few recent studies have examined the prevalence of vitamin D deficiency in the elderly Spanish population, although all agree on its high prevalence in this age group. One study

found that 33.9% of the total Spanish population might be at risk of vitamin D deficiency [25] while others report a prevalence of up to 80% in people living in nursing homes [9, 10].

Given the high prevalence of vitamin D deficiency in older adults, and the results of this study that show the course of SARS-CoV-2 to be more serious in patients with this deficiency, vitamin D supplementation might recommendable for the elderly, especially those who are institutionalised or who are walking dependent.

Some differences were seen between the three sub-periods analysed. The first, from March to June 2020, registered the largest number of patients with SARS-CoV-2 infection, while the third, from January to March 2021, registered the highest prevalence of SARS-CoV-2 infection with vitamin D deficiency. Overall, SARS-CoV-2 infections declined over time, perhaps due to changes in viral virulence (e.g., an Alpha variant first detected in Great Britain was in circulation during the third sub-period), better knowledge of the disease and more efficient treatments, or the influence of the season. Certainly, an increased prevalence of SARS-CoV-2 with vitamin D deficiency was recorded during the winter, similar to that reported by other authors who suggest the association between vitamin D levels and severity of infection may be stronger during this season when this vitamin deficiency is generally more common [17].

The main strength of the present study is its large sample of real-world data. It does, however, suffer from certain limitations. For example, unlike in classical research, reproducibility is not generally taken into account in big-data studies since these involve large amounts of information collected from the whole target population. Because the data analysed in the present work was exclusively captured from EHRs, the quality of the results reported for the different variables is necessarily tied to that of the clinical records themselves; EHRs can be partially incomplete and may not capture all relevant clinical information. Finally, the study sample involved the records of patients with SARS-CoV-2 confirmed by PCR or an antigen test plus clinical and epidemiological suspicion; it is therefore possible that some false positive cases were included in the final study sample.

## Conclusions

A significant association would appear to exist between vitamin D deficiency and a more serious course of SARS-CoV-2 infection, including an increased risk of hospitalisation. Institutionalised and dependent people aged over 65 years are more commonly registered among patients with SARS-CoV-2 plus vitamin D deficiency, although more research is necessary to draw a definitive conclusion on any causal link.

## Acknowledgments

The authors thank Eduardo Cañada of the Computer Dept., Infanta Sofía University Hospital, for assistance with informatic techniques and also want to thank Adrian Burton for his considered English review to improve the paper.

## Author Contributions

**Conceptualization:** Marta Neira Álvarez, Gema Navarro Jiménez, Noemi Anguita Sánchez, Mª del Mar Bermejo Olano, Carmen Sáez Nieto.

**Formal analysis:** Marta Neira Álvarez, Mª del Mar Bermejo Olano, Rocio Queipo, María Benavent Núñez.

**Investigation:** Marta Neira Álvarez, Rocio Queipo, María Benavent Núñez, Alejandro Parralejo Jimenez, Guillermo López Yepes.

**Methodology:** Marta Neira Álvarez, Noemi Anguita Sánchez, Rocio Queipo.

**Resources:** Noemi Anguita Sánchez, María Benavent Núñez, Alejandro Parralejo Jimenez, Guillermo López Yepes.

**Supervision:** Noemi Anguita Sánchez, Rocio Queipo.

**Validation:** Noemi Anguita Sánchez.

**Writing – original draft:** Gema Navarro Jiménez.

**Writing – review & editing:** Noemi Anguita Sánchez.

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
