## [Decision Letter · Decision Letter 0]

2 Dec 2022

PONE-D-22-29455Vitamin D deficiency and SARS‑CoV‑2 infection: Big-data analysis from March 2020 to March 2021. D-COVID studyPLOS ONE

Dear Dr. Anguita,

Thank you for submitting your manuscript to PLOS ONE. After careful consideration, we feel that it has merit but does not fully meet PLOS ONE’s publication criteria as it currently stands. Therefore, we invite you to submit a revised version of the manuscript that addresses the points raised during the review process.

We look forward to receiving your revised manuscript.

Kind regards,

Ibrahim Umar Garzali, MBBS, FWACS

Academic Editor

PLOS ONE

Journal Requirements:

a) Did participants provide their written or verbal informed consent to participate in this study?

4. Please amend your authorship list in your manuscript file to include author "Noemi Anguita". 

5. Please amend the manuscript submission data (via Edit Submission) to include authors: Neira Álvarez, Navarro Jiménez, Anguita Sánchez N, Bermejo Olano M.M, Queipó R, Benavent Nuñez M, Parralejo Jimenez A, López Yepes G, Sáez Nieto C. 

6. One of the noted authors is a group or consortium [Neira Álvarez M a, Navarro Jiménez G b , Anguita Sánchez N C*, Bermejo Olano M.M c, Queipó R d, Benavent Nuñez M e, Parralejo Jimenez A e, López Yepes G e, Sáez Nieto C f]. In addition to naming the author group, please list the individual authors and affiliations within this group in the acknowledgments section of your manuscript. Please also indicate clearly a lead author for this group along with a contact email address.

7. Please include a separate caption for figure in your manuscript.

Reviewers' comments:

Reviewer's Responses to Questions

**Comments to the Author**

1. Is the manuscript technically sound, and do the data support the conclusions?

Reviewer #1: No

Reviewer #2: No

Reviewer #3: Yes

Reviewer #4: Yes

Reviewer #5: Yes

Reviewer #6: Partly

Reviewer #7: Partly

2. Has the statistical analysis been performed appropriately and rigorously? 

Reviewer #1: No

Reviewer #2: No

Reviewer #3: Yes

Reviewer #4: Yes

Reviewer #5: Yes

Reviewer #6: Yes

Reviewer #7: No

3. Have the authors made all data underlying the findings in their manuscript fully available?

Reviewer #1: Yes

Reviewer #2: Yes

Reviewer #3: Yes

Reviewer #4: Yes

Reviewer #5: Yes

Reviewer #6: Yes

Reviewer #7: Yes

4. Is the manuscript presented in an intelligible fashion and written in standard English?

Reviewer #1: No

Reviewer #2: No

Reviewer #3: Yes

Reviewer #4: Yes

Reviewer #5: No

Reviewer #6: No

Reviewer #7: No

5. Review Comments to the Author

Reviewer #1: General comments

1. Use standard English spellings. Example- ‘terciary’ should be ‘tertiary’

2. Several grammatical errors throughout the manuscript. English language flow should be improved.

3. Spell out all the abbreviations when first used in the manuscript, including COVID-19, SAVANA, EHR and others. Do not use abbreviations in the abstract.

Specific Comments

1. I do not understand why the ethical approval was taken from University Hospital Getafe, while the study was performed in University Hospital Madrid. Are they same institutions or different?

2. There are several studies done throughout the world studying relationship between vitamin D level and severity of covid-19. Though authors have taken examples of studies which have shown significant relationship between vitamin D level and covid-19 severity, there are other studies which have not shown any relationship between the two. There is no mention of such studies in introduction and discussion. The manuscript needs to have balanced unbiased discussion of all positive as well as negative studies.

3. There was a large cohort study done in UK (Biobank study) with more than 3 hundred thousand participants, among them about 500 developed covid-19 but there was no association between vitamin D level and risk or severity of covid 19. However this study is only a cross-sectional study, which is not an ideal risk factor study design.

4. As we can see from the result, those with vitamin D deficiency were elderly than those with normal vitamin D. Also the deficient patients were more debilitated and suffered chronic illnesses. The age, chronic diseases are themselves independent risk factors for developing covid-19 severity and increased death. Hence they are the confounding variables. Analysis has been done without adjusting for these confounding variables. Hence the current result showing association cannot be validated.

5. Moving forward from what this study has attempted to establish, several intervention studies have already been performed comparing the effects of vitamin D supplementation versus no supplementation, in covid-19 patients with vitamin D deficiency. But no studies have so far established that vitamin D supplementation reduces severity of covid-19 in patients with deficiency. One such study was done in Spain which showed reduced ICU admission among supplemented patients, but sample size was lower and non-supplemented patients had more chronic diseases like hypertension and DM (hence inadequate matching). One large trial was done in Norway with 34000 patients, but failed to show any benefit with vitamin D supplementation. In conclusion, based on trial results till date, vitamin D deficiency cannot be ascertained as risk factor for severity of covid-19. This should be acknowledged by the authors in their manuscript.

Reviewer #2: Authors are careless even to not know the simple differences between a comma and full stop (especially in the abstracts and tables) where they claim 143.157 analyzed patients.

Major confounding factors is age and comorbidities like > 80years, nursing home residents are not eliminated in the study which is higher in the vitamin D deficient groups which can lead to increased mortality. When these baseline confounding factors are matched true effects of vitamin D deficiency can be seen. We need to compare > 80years vitamin D deficient with normal Vitamin D levels and so on. All the confounding factors should be compared between Vitamin D deficient and normal Vitamin D group should be available for the results.

Also, including outpatient and ICU admitted patients cannot be compared and included in one pool to make a final results as they are in two extremes of disease course and progression.

Reviewer #3: The title of the manuscript should contain the type of the study. I think this study was better if it was done as multicenter research to get more appropriate results however current situation doesn't weaken the study

Reviewer #4: The author has used big data with a novel methodological approach, I think this is an interesting study. Some suggestions for authors:

1. There are some words that are still spelled in Spanish, and some words that lack letters, this must be corrected.

2. There are 4 phases of the data extraction process, I think the validation process in the fourth phase should be explained more about how to validate it.

3. There is no discussion regarding the differences in the severity of Covid-19 in three different periods related to vitamin D levels. The author should add this to the discussion.

4. The inclusion criteria in this study were patients who were confirmed positive for Covid from the PCR and antigen tests, but why did the authors add clinical suspicion and epidemiological circumstances to the study's weaknesses?

Reviewer #5: The manuscript require editing in terms of grammar and spelling.

There should be specific reference in regards whether vitamin D levels were collected during the phase of infection or baseline levels prior to infection. In addition, are those patients on vitamin D supplementation included in the study or excluded.

Reviewer #6: The authors address a relevant theme regarding the theme of vitamin D and its immunomodulatory role. Involving this issue in the pandemic theme by COVID-19 is timely and relevant. Some methodological and writing aspects deserve to be considered in order to improve the manuscript.

Abstract: The authors make a brief overview of the theme and objectives. However, in the methodology, they do not present the study design or the definitions that guide the analysis of the outcomes. This makes understanding the results confusing. I suggest reediting this field to give the reader a better understanding of what has been studied and found.

Introduction. In the introduction, the authors are direct to the point under analysis and present the objectives in a clear and detailed way. No further comment at this point.

Methods.

1. The authors describe the variables, covariable, outcome and outcome measures on a good way. However, on page 4, they present EHR. I recommend that the expression of the acronym be expressed in full and, later, as it is reproduced in the text, use it separately.

2. The authors present the variables separated by "/". I recommend that you review the grammar and separate them by comma, as suggested by grammar.

3. The variable "Time of episode" is unclear. I suggest writing the definition of this variable and how it was measured in the study.

Results and discussion. The results and discussion are presented with consistent and organized data. However, as the methodology needs to be improved, the accuracy of the results and discussion suffers from the dependence on the quality of the methodology section.

Discussion. It is not written in English. Please, revise it.

Reviewer #7: This retrospective study evaluated the frequency of Vit D deficiency in a Sars CoV-2 infected population while unravelling the clinical course of COVID-19 in patients with hypovitaminosis-D across three periods. My comments are below

1. How many patients were excluded due to incomplete data?

2. It is tricky to attribute the clinical course and outcome of the study - long hospital stay, ICU admission, increased morbidity, and mortality solely to Hypovitaminosis D, because the cohort with Hypovitaminosis D also have various confounders such as malnutrition, habitation in nursing home, Obesity, HTA in addition to increased age. A multivariate regression analysis, correcting for these cofounders is necessary to determine if indeed Hypovitaminosis-D accounted for these clinical courses.

3. The age- and gender-adjusted rate and OR of Hypovitaminosis- D should be provided considering that the condition is influenced by age and gender.

4. All abbreviations should be defined at first mention – for example, I did not see the full definition of SAVANA. Is this an abbreviation?

5. The discussion should be strengthened. A chunk of the discussion is a repetition of the results. The translational value of the findings of the study and its implication for COVID management in patients with Hypovitaminosis-D should be discussed.

6. Are there factors such weather and climate factors, human factors and viral virulence factors that could have influenced the clinical course of COVID-19 infection in the three periods under consideration. These should be discussed as well. For example, is the same Sars- Cov2 variant responsible for the infection during the three periods under review?

7. The article deserves to be rewritten by a native English speaker for easy readability.

6. PLOS authors have the option to publish the peer review history of their article (what does this mean?). If published, this will include your full peer review and any attached files.

Reviewer #1: **Yes: **Sudeep Adhikari

Reviewer #2: **Yes: **Avinash Adiga

Reviewer #3: No

Reviewer #4: No

Reviewer #5: No

Reviewer #6: No

Reviewer #7: **Yes: **Philip B Adebayo

---

## [Author Response · Author response to Decision Letter 0]

22 May 2023

Response letter to Reviewers’ Comments 

We would like to thank the reviewers for their time and dedication to read our paper, as well as the valuable comments provided. Below you can find a point-by-point response to the reviewer’s comments. We formatted the article according to the requirements of the journal. In addition, we have made the changes in the word document, clearly highlighting them (track-system). We have reviewed the English spelling and grammar as recommended using an external company (www.physicalevidence.es).

Thanks in advance.

All authors.

Reviewer 1

General comments

1. Use standard English spellings. Example- ‘terciary’ should be ‘tertiary’

2. Several grammatical errors throughout the manuscript. English language flow should be improved.

3. Spell out all the abbreviations when first used in the manuscript, including COVID-19, SAVANA, EHR and others. Do not use abbreviations in the abstract.

We have reviewed the English spelling and grammar as recommended.

We have also checked all the abbreviations when first used in the manuscript

Specific Comments

1. I do not understand why the ethical approval was taken from University Hospital Getafe, while the study was performed in University Hospital Madrid. Are they same institutions or different? 

In Madrid there are a limit number of Research and Ethical Comitee and Hospital Infanta Sofia has no Ethical Comitee for Research so we have to ask for evaluation and approval to other reference instutions. Getafe University Hospital is one of the reference institutions to ask for it. This is a usual practice as it is the only way for Ethical aprproval.

2. There are several studies done throughout the world studying relationship between vitamin D level and severity of covid-19. Though authors have taken examples of studies which have shown significant relationship between vitamin D level and covid-19 severity, there are other studies which have not shown any relationship between the two. There is no mention of such studies in introduction and discussion. The manuscript needs to have balanced unbiased discussion of all positive as well as negative studies.

Thank you very much for these observations; we have introduced all these aspects to balance discussion. We have made changes in Introduction Section and Discussion. See below (point 5)

3. There was a large cohort study done in UK (Biobank study) with more than 3 hundred thousand participants, among them about 500 developed covid-19 but there was no association between vitamin D level and risk or severity of covid 19. However this study is only a cross-sectional study, which is not an ideal risk factor study design.

We have introduced the results of this study (see bellow)

4. As we can see from the result, those with vitamin D deficiency were elderly than those with normal vitamin D. Also the deficient patients were more debilitated and suffered chronic illnesses. The age, chronic diseases are themselves independent risk factors for developing covid-19 severity and increased death. Hence they are the confounding variables. Analysis has been done without adjusting for these confounding variables. Hence the current result showing association cannot be validated.

Thank you very much for this analysis,; we have done three multiple logistic regression to assess differences in mortality, ICU admission and hospitalitation rates between groups (with and without vitamin D deficiency) adjusting by covariates (age, obesity, malnutrition, living in nursing home and SARS-CoV- 2 infection). The results were expressed as odds ratios (OR) with 95% CI. As you mentioned confounding variables have a great influence in Covid-19 evolution and consequently we have changed discusion and conclusión.

5. Moving forward from what this study has attempted to establish, several intervention studies have already been performed comparing the effects of vitamin D supplementation versus no supplementation, in covid-19 patients with vitamin D deficiency. But no studies have so far established that vitamin D supplementation reduces severity of covid-19 in patients with deficiency. One such study was done in Spain which showed reduced ICU admission among supplemented patients, but sample size was lower and non-supplemented patients had more chronic diseases like hypertension and DM (hence inadequate matching). One large trial was done in Norway with 34000 patients, but failed to show any benefit with vitamin D supplementation. In conclusion, based on trial results till date, vitamin D deficiency cannot be ascertained as risk factor for severity of covid-19. This should be acknowledged by the authors in their manuscript.

Thank you very much for these observations; we have introduce all these aspects to balance discussion. We have made changes in Introduction Section and Discussion. See below:

Introduction: “However, other authors report inconsistent results regarding vitamin D status and diagnosis with SARS-CoV-2 infection, the lack of any association with hospitalization or mortality, and no difference in the length of hospital stay when patients with moderate to severe SARS-CoV-2 infection are provided with oral supplements of vitamin D. The evidence supporting a role for vitamin D in SARS-CoV-2 infection therefore remains inconclusive..”

Discussion: A recent meta-analysis examined the relationship between SARS-CoV-2 infection and vitamin D status. Eight retrospective studies were found that investigated the association between vitamin D levels and SARS-CoV-2 infection, all of which reported that subjects with low vitamin D levels had double the risk. However, most of these studies included unadjusted estimates, rendering it impossible to assess these characteristics in terms of other variables. In addition, 16 studies investigated the association between vitamin D levels and the severity of SARS-CoV-2 infection, reporting double the risk for subjects with low vitamin D levels, even when adjusting for confounding factors (comorbidity, age, BMI and sex). Moreover, supplementation was associated with a significantly lower risk of severe disease in six studies, and with a lower risk of mortality in eight. (D’Ecclesiis O, et al)

Other authors have reported similar results for a Spanish population, noting an increased risk of hospital admission and need for critical care among those with low vitamin D levels, but no increase in mortality. Another retrospective study reported an inverse relationship between vitamin D levels and length of hospital stay for SARS-CoV-2 infection. 

The relationship between vitamin D and SARS-CoV-2 infection was initially raised because, prior to the pandemic, a significant association was reported between low serum 25(OH) vitamin D levels and the severity of acute respiratory tract infections in both adults and children2, , . Certainly, vitamin D modulates angiotensin receptor expression in the lungs, stimulates pulmonary surfactant production, reduces hyperinflammatory cytokine storms, and increases levels of regulatory T lymphocytes, which together might explain why vitamin D reduces the incidence of acute respiratory infection and the severity of respiratory tract disease (repeat ref 25). Vitamin D also has important protective effects on the cardiovascular system, including the improvement of myocardial contractility and anti-thrombotic effects. Moreover, vitamin D deficiency has been associated with diabetes, hypertension and dyslipidemia. It may be, therefore, that this vitamin protects against cardiovascular complications in patients with SARS-CoV-2 infection . Nevertheless, two studies failed to find any protective effects of vitamin D supplements when treating patients hospitalised with SARS-CoV-2, although in both studies the sample was small and heterogeneous, and only patients with severe disease were included , . 

Reviewer 2

Authors are careless even to not know the simple differences between a comma and full stop (especially in the abstracts and tables) where they claim 143.157 analyzed patients.

We are very sorry about these mistakes. Thank you for this observation; we have review all the figures numbers

Major confounding factors is age and comorbidities like > 80years, nursing home residents are not eliminated in the study which is higher in the vitamin D deficient groups which can lead to increased mortality. When these baseline confounding factors are matched true effects of vitamin D deficiency can be seen. 

We need to compare > 80years vitamin D deficient with normal Vitamin D levels and so on. All the confounding factors should be compared between Vitamin D deficient and normal Vitamin D group should be available for the results.

Also, including outpatient and ICU admitted patients cannot be compared and included in one pool to make a final results as they are in two extremes of disease course and progression.

Thank you very much for this analysis; we have done three multiple logistic regression to assess differences in mortality, ICU admission and hospitalitation rates between groups (with and without vitamin D deficiency) adjusting by covariates (age, obesity, malnutrition, living in nursing home and SARS-CoV- 2 infection). The results were expressed as odds ratios (OR) with 95% CI. As you mentioned confounding variables have a great influence in Covid-19 evolution and consequently we have changed discusion and conclusión.

We also are aware that Covid-19 results in a broad spectrum of disease, with great differences in number of cases, severity and evolution and including more than 80% of patients showing few or no symptoms. And that is why we only have included patients with sintomatic disease so they went to Emergency Dept at Hospital. Patients with few or no sintoms were treated at Primary Care so they are not included in this study.

Reviewer 3

The title of the manuscript should contain the type of the study.

We have included type of study: 

“Vitamin D deficiency and SARS‑CoV‑2 infection: a retrospective case–control study with big-data analysis covering March 2020 to March 2021.”

 I think this study was better if it was done as multicenter research to get more appropriate results however current situation doesn't weaken the study

Big Data anlysis requires a special methodology that is not included in most of public hospitals in Madrid so it was a real oportunity to do it in our hospital. We hope in future big data studies will be multicenter research.

Reviewer 4

The author has used big data with a novel methodological approach, I think this is an interesting study. Some suggestions for authors:

1. There are some words that are still spelled in Spanish, and some words that lack letters, this must be corrected.

We have done a deep review of English style and gramar 

2. There are 4 phases of the data extraction process, I think the validation process in the fourth phase should be explained more about how to validate it. SAVANA

We have done some chages to clarify validation process See below:

“4) Validation. This entails verifying the system’s accuracy in identifying records that contain mentions of vitamin D deficiency, SARS-CoV-2 and related variables. For this, a human-annotated corpus of EHRs known as an “annotation gold standard” is created for comparison. In the present work the SampLe Calculator for Evaluation (SLiCE) tool was used to determine the size of the corpus required (with a confidence level of 95% and interval widths of 5%) based on the prevalence of mentions of the primary research variables (vitamin D deficiency, SARS-CoV-2). Although the number of EHRs required to adequately capture linguistic occurrences in order to generate reliable performance measurements is still a matter of debate, this methodology provides a robust estimate of precision (P) and recall (R) values. The annotation gold standard was constructed using 98 records annotated by two designated researchers ("the annotators") using the Savana Evaluation Tool to manually correct the EHRead-selected variables. The Inter-Annotator Agreement (IAA) was determined using the F1-Score. A third researcher resolved the differences when the annotators disagreed. 

The generated gold standard corpus was then used to assess performance of the EHRead technology, calculating:

• Precision = tp/(tp+fp). This shows how well the system recalls important therapeutic concepts.

• Recall = tp/(tp+fn). This represents the volume of data that the system retrieves.

• F1-Score= (2 x precision x recall/(precision + recall). This provides an indication of retrieval performance.

In each case, tp is the total number of true positives (i.e., correctly retrieved records), fn is the total number of false negatives (i.e., wrongly omitted records), and fp is the total number of false positives (i.e., records incorrectly retrieved) . 

The linguistic evaluation of the variable 'SARS-CoV-2' yielded accuracy, recall and F-scores of 0.61, 0.87, and 0.72, respectively. The Interannotator Agreement was 0.70. Other primary variables such as hypovitaminosis D and malnutrition yielded F-scores of >0.70.”

3. There is no discussion regarding the differences in the severity of Covid-19 in three different periods related to vitamin D levels. The author should add this to the discussion.

We have introduce some discusión about Covid-19 and vitamin D deficiency in the different periods of pandemia.

“ Some differences were seen between the three sub-periods analysed. The first, from March to June 2020, registered the largest number of patients with SARS-CoV-2 infection, while the third, from January to March 2021, registered the highest prevalence of SARS-CoV-2 infection with vitamin D deficiency. Overall, SARS-CoV-2 infections declined over time, perhaps due to changes in viral virulence (e.g., an Alpha variant first detected in Great Britain was in circulation during the third sub-period), better knowledge of the disease and more efficient treatments, or the influence of the season. Certainly, an increased prevalence of SARS-CoV-2 with vitamin D deficiency was recorded during the winter, similar to that reported by other authors who suggest the association between vitamin D levels and severity of infection may be stronger during this season when this vitamin deficiency is generally more common (D’Ecclesiis O, Gavioli C, Martinoli C, Raimondi S, Chiocca S, Miccolo C, et al. (2022) Vitamin D and SARS-CoV2 infection, severity and mortality: A systematic review and meta-analysis. PLoS ONE 17(7): e0268396. https://doi.org/10.1371/journal.pone.0268396)”

4. The inclusion criteria in this study were patients who were confirmed positive for Covid from the PCR and antigen tests, but why did the authors add clinical suspicion and epidemiological circumstances to the study's weaknesses?

During pandemic period, diagnosis of Covid-19 infection in our hospital was made by PCR positive test or when there was high clinical suspician and positive antigenic test (under these circunstances antigenic test has high sensitivity and spcificity: 90% and 97 % respectivly) so we have consider that there are low but possible false positives and that is why have added to weaknesses.

Reviewer 5

The manuscript require editing in terms of grammar and spelling.

We have done a deep review of English style and gramar again.

There should be specific reference in regards whether vitamin D levels were collected during the phase of infection or baseline levels prior to infection. In addition, are those patients on vitamin D supplementation included in the study or excluded.

We have introduced information in Materials and Methods (Study variables) about these two questions: 

-Vitamin D deficiency: considering those subjects with a diagnosis of vitamin D deficiency, hypovitaminosis or when vitamin D blood levels were < 30mg/dl described in th EHR before or during the SARS-CoV-2 infection.

-Vitamin D treatments: if the patient was on vitamin D supplementation before or during the SARS-CoV- 2 infection.

Reviewer 6

The authors address a relevant theme regarding the theme of vitamin D and its immunomodulatory role. Involving this issue in the pandemic theme by COVID-19 is timely and relevant. Some methodological and writing aspects deserve to be considered in order to improve the manuscript.

Abstract: The authors make a brief overview of the theme and objectives. However, in the methodology, they do not present the study design or the definitions that guide the analysis of the outcomes. This makes understanding the results confusing. I suggest reediting this field to give the reader a better understanding of what has been studied and found.

Thank you for this observation. We have reorganized this sections as requested in order to improve the clarity and flow.

Introduction. In the introduction, the authors are direct to the point under analysis and present the objectives in a clear and detailed way. No further comment at this point.

Thank you very much for your comments

Methods.

1. The authors describe the variables, covariable, outcome and outcome measures on a good way. However, on page 4, they present EHR. I recommend that the expression of the acronym be expressed in full and, later, as it is reproduced in the text, use it separately.

Thanks. We have also checked all the abbreviations when first used in the manuscript; EHR means electronic health records and it has been added the meaning and acronym.

2. The authors present the variables separated by "/". I recommend that you review the grammar and separate them by comma, as suggested by grammar.

We have changed “/” by comma in the manuscript.

3. The variable "Time of episode" is unclear. I suggest writing the definition of this variable and how it was measured in the study.

We have added some information to clarify “Time of episode: date, month and year of SARS-CoV- 2 infection in order to clasify in one of the three periods: from March 2020 to June 2020, from August 2020 to October 2020 or from January 2021 to March 2021.”

Results and discussion. The results and discussion are presented with consistent and organized data. However, as the methodology needs to be improved, the accuracy of the results and discussion suffers from the dependence on the quality of the methodology section.

Thank you very much. We have done a new Materials and Methods section following your suggestions with different subheadings.

Discussion. It is not written in English. Please, revise it.

We have done a deep review of English style and gramar using an external service with a native English speaker.

Reviewer 7

This retrospective study evaluated the frequency of Vit D deficiency in a Sars CoV-2 infected population while unravelling the clinical course of COVID-19 in patients with hypovitaminosis-D across three periods. My comments are below

1. How many patients were excluded due to incomplete data? SAVANA

The information of qualitative variables that do not appear in the medical record has been taken as informative. Qualitative clinical variables that do not appear in the electronic health records were considered to be negated.

As for quantitative variables, they have not been taken into account for the mean in the case of absence of the variable (example: mean stay in Intensive Care Unit).

We do not exclude patients even though the complete information is not available for each variable; only those patients in whom there was an error in the date of birth were excluded as age was considered a minimum necessary record (less than 1 day, older than 120 years or age absent). In this sense, 6666 patients were excluded. We have included this information in paper.

2. It is tricky to attribute the clinical course and outcome of the study - long hospital stay, ICU admission, increased morbidity, and mortality solely to Hypovitaminosis D, because the cohort with Hypovitaminosis D also have various confounders such as malnutrition, habitation in nursing home, Obesity, HTA in addition to increased age. A multivariate regression analysis, correcting for these cofounders is necessary to determine if indeed Hypovitaminosis-D accounted for these clinical courses.

Thank you very much for these observations. Please see bellow that we have answered to this question.

3. The age- and gender-adjusted rate and OR of Hypovitaminosis- D should be provided considering that the condition is influenced by age and gender.

Thank you very much for this analysis; we have done multiple logistic regression to assess differences in mortality, ICU admission and hospitalitation rates between groups (with and without vitamin D deficiency) adjusting by covariates (age, obesity, malnutrition, living in nursing home and SARS-CoV- 2 infection). The results were expressed as odds ratios (OR) with 95% CI. As you mentioned confounding variables have a great influence in Covid-19 evolution and consequently we have changed discusion and conclusión.

4. All abbreviations should be defined at first mention – for example, I did not see the full definition of SAVANA. Is this an abbreviation?

Savana is an international medical company that provided the technology for the extraction oif clinical variables from EHRs using NLP and ML (https://savanamed.com/ )

5. The discussion should be strengthened. A chunk of the discussion is a repetition of the results. The translational value of the findings of the study and its implication for COVID management in patients with Hypovitaminosis-D should be discussed.

Thank you very much for the observation we have done a deep review of the discusión and introduced new ideas and aspects about results.

6. Are there factors such weather and climate factors, human factors and viral virulence factors that could have influenced the clinical course of COVID-19 infection in the three periods under consideration. These should be discussed as well. For example, is the same Sars- Cov2 variant responsible for the infection during the three periods under review?

We have discussed differences between the three periods of time and try to find new aspects regarding differences. See below changes

“ Some differences were seen between the three sub-periods analysed. The first, from March to June 2020, registered the largest number of patients with SARS-CoV-2 infection, while the third, from January to March 2021, registered the highest prevalence of SARS-CoV-2 infection with vitamin D deficiency. Overall, SARS-CoV-2 infections declined over time, perhaps due to changes in viral virulence (e.g., an Alpha variant first detected in Great Britain was in circulation during the third sub-period), better knowledge of the disease and more efficient treatments, or the influence of the season. Certainly, an increased prevalence of SARS-CoV-2 with vitamin D deficiency was recorded during the winter, similar to that reported by other authors who suggest the association between vitamin D levels and severity of infection may be stronger during this season when this vitamin deficiency is generally more common (D’Ecclesiis O, Gavioli C, Martinoli C, Raimondi S, Chiocca S, Miccolo C, et al. (2022) Vitamin D and SARS-CoV2 infection, severity and mortality: A systematic review and meta-analysis. PLoS ONE 17(7): e0268396. https://doi.org/10.1371/journal.pone.0268396)”

7. The article deserves to be rewritten by a native English speaker for easy readability.

We have done a deep review of English style and gramar using an external company for this purpose.

---

## [Editor Report · Decision Letter 1]

25 May 2023

Vitamin D deficiency and SARS‑CoV‑2 infection: a retrospective case–control study with big-data analysis covering March 2020 to March 2021

PONE-D-22-29455R1

Dear Dr. Anguita,

We’re pleased to inform you that your manuscript has been judged scientifically suitable for publication and will be formally accepted for publication once it meets all outstanding technical requirements.

Kind regards,

Ibrahim Umar Garzali, MBBS, FWACS

Academic Editor

PLOS ONE
---

## [Editor Report · Acceptance letter]

29 Aug 2023

PONE-D-22-29455R1 

Vitamin D deficiency and SARS‑CoV‑2 infection: a retrospective case–control study with big-data analysis covering March 2020 to March 2021 

Dear Dr. Anguita:

I'm pleased to inform you that your manuscript has been deemed suitable for publication in PLOS ONE. Congratulations! Your manuscript is now with our production department. 

Kind regards, 

on behalf of

Dr. Ibrahim Umar Garzali 

Academic Editor

PLOS ONE